# Evolution of the Composition and Melting Behavior of Spinnable Pitch during Incubation

**DOI:** 10.3390/molecules28031097

**Published:** 2023-01-21

**Authors:** Qian Li, Pingping Zuo, Shijie Qu, Wenzhong Shen

**Affiliations:** 1State Key Laboratory of Coal Conversion, Institute of Coal Chemistry, Chinese Academy of Sciences, 27 South Taoyuan Road, Taiyuan 030001, China; 2Center of Materials Science and Optoelectronics Engineering, University of Chinese Academy of Sciences, No. 19 (A) Yuquan Road, Shijingshan District, Beijing 100049, China

**Keywords:** spinnable pitch, fraction separation, chemical components, rheological properties

## Abstract

The physical and chemical properties of spinnable pitch showed a huge impact on the performance of resultant pitch carbon fiber even if its physical and chemical properties were slightly changed. Various polycyclic aromatic compounds and abundant free radicals existed in spinnable pitch, and there are many interactions among molecules and free radicals. The molecular structure and composition of spinnable pitch were investigated during incubation, and the effect of molecular evaluation on rheological properties of spinnable pitch was illustrated using various characterization methods in this work. It indicated that n-hexane soluble fraction mainly occurred condensation or cleavage, and a small number of heavy components were generated after a long period. The fraction of n-hexane insoluble/toluene soluble underwent molecular condensation and cross-linking in the presence of oxygen-containing radicals and aromatic hydrocarbon radicals, while toluene insoluble/tetrahydrofuran soluble fraction tended to change in large molecules of polycyclic aromatic hydrocarbons. Lastly, tetrahydrofuran insoluble fraction was condensed due to its high aromaticity during the incubation process, and the content of aromatic carbon increased. These changes of composition and structure of spinnable pitch led to its softening point, increase in viscosity and flow activation energy, and deterioration of the rheological property.

## 1. Introduction

As the by-product of coal coking, coal tar pitch is a complex mixture system composed of a variety of polycyclic aromatic hydrocarbons, which is prone to thermal condensation or crosslinking [1]. Coal tar pitch is widely selected to produce functional carbon materials due to its easy graphitization and high carbon yield [2,3], such as needle coke [4], graphite electrode/anode material [5], pitch-based carbon fiber [6,7], porous carbon, etc. As the precursor of pitch-based carbon fiber, spinnable pitch derived from coal tar pitch needs to meet the requirements of low ash content and quinoline insoluble, relatively concentrated molecular weight distribution, good rheological properties, high softening point, and wide spinning temperature range [8,9]. However, the spinnable pitch has a complex molecular structure with a high degree of condensation and high aromaticity. These components are varied greatly with coal type and source, pyrolysis, and separation process. There are several interaction modes with different strengths, such as acid-base interaction, hydrogen bond, π−π interaction, van der Waals force, and electrostatic interaction [10,11,12], and it is easy to form different micelles and dispersion systems. At present, the reported analysis methods can only recognize and speculate on the spinnable pitch to a certain extent [3,13,14], and cannot accurately describe its composition and structure, let alone provide effective guidance for its composition and structure control.

As a complex aromatic mixture system and a variety of molecular interactions, the microstructure and composition of the spinnable pitch will change with the extension of time during incubation, which will change the rheological properties and spinning properties of spinnable pitch and ultimately affect the performance of the pitch-based carbon fiber. Some new unexpected phenomena occurred during spinnable pitch preparation. It is difficult to remove the soluble components in spinnable pitch by multiple extractions with the same solvent. After melting and cooling cycle, the softening point, molecular weight distribution, and aromaticity of the spinnable pitch do not change, but its spinnability and liquid phase carbonization performance fluctuate greatly. Spinnable pitch and mesophase pitch with the same chemical composition, molecular weight distribution, and softening point prepared from the same coal pitch show great differences in rheological viscosity-temperature characteristics and fiber-forming behavior. After the same heat treatment process, the physical and chemical properties of the final pitch carbon fiber are greatly different.

These phenomena showed that it was far from enough to improve the performance of spinnable pitch only by apparent parameters such as softening point, molecular weight distribution, quinoline insolubles, and carbon yield. It is necessary to further understand the fraction structure and evolution of spinnable pitch from the perspective of component distribution, molecular existing state, and their association at the molecular level based on the concept and technical methods of molecular chemical engineering.

In general, the microscopic properties of spinnable pitch molecules are mainly studied from the aspects of elemental composition, functional groups, molecular weight, melting behavior, etc. It was reported that the saturated and aromatic components were partially converted to colloid and asphaltene after 50 h of thin film oven test [15], and an oxidized pitch model was established by the molecular dynamics method [16]. The relationship between nano-morphologic parameters and properties of the pitch in the process of aging was investigated by atomic mechanics microscope to establish a functional relationship model between nano-morphologic parameters and properties of pitch [17]. The contents of carbonyl and sulfoxide groups in pitch were gradually increased after performing a thin-film oven test for 85 min and a pressure aging vessel for 40 h [18]; the ratio of large molecular components in pitch increased after aging at 165 °C for 5 h [19]. These results provided a reference for the performance changes of the spinnable pitch during incubation, but it was necessary to further understand the structural evolution in the molecular composition, distribution, rheological property, and fiber-forming behavior of spinnable pitch, to provide a basis for the regulation of the physical and chemical properties of the spinnable pitch.

Here, four fractions in spinnable pitch were separated and characterized to illustrate the change of chemical component during incubation. This showed that oxygen took part in the reaction and produced oxygen-containing functional groups in each fraction. Pitch molecules underwent oxidation, cross-linking, condensation, and cleavage processes during incubation. The viscosity, flow activation energy, and softening point of spinnable pitch were increased, while its rheological property was deteriorated; thus, the spinnable performance was worse. These results are helpful to understand the composition evolution of spinnable pitch and its influence on the spinning performance of pitch fiber, and provide a new perspective for the preparation of pitch carbon fiber.

## 2. Results and Discussions

### 2.1. Extraction Separation and Fraction Distribution

In order to investigate the changes of the spinnable pitch with incubation time in more detail, the spinnable pitch was extracted and divided into n-hexane soluble (HS), n-hexane insoluble/toluene soluble (HI-TS), toluene insoluble/tetrahydrofuran soluble (TI-THFS), and tetrahydrofuran insoluble (THFI) fractions by solvent extraction; the separation procedure is shown in Figure 1.

Table 1 lists the different fraction contents of spinnable pitch with incubation time (0 to 150 days). TI-THFS had the largest change range and THFI had the smallest change range. The change of solubility of the spinnable pitch during incubation indicated that its components had also changed. Figure 1 shows the molecular distribution of the whole fraction of spinnable pitch, HI-TS, TI-THFS, and THFI fractions before and after incubation. The distribution of the molecular weight of pitch did not change significantly after incubation, but the peak intensity decreased. The molecular weight distribution of HI-TS, TI-THFS, and THFI fractions was gradually increased [20]. The molecular weight of the THFI fraction was larger than that both of TI-THFS and HI-TS. This indicated that the component changed, and the degree of conjugation of the molecules of pitch decreased after incubation.

The elemental composition of spinnable pitches at different incubation times are listed in Table 2; spinnable pitch had a high carbon content, of about 93%, and a hydrogen content of about 4%. The oxygen content gradually increased with time. The nitrogen and sulfur contents were less than 1% and without any significant change in all samples, while the H/C molar ratio decreased and O/C increased.

### 2.2. HS Fraction Variation

The molecular weight distribution in HS fraction of spinnable pitch was investigated by gel permeation chromatography (GPC). The molecular weight is shown in Figure 2. The average molecular weight and the number average molecular weight during the incubation are listed in Table 3. This suggested that the molecular weight distribution of the HS fraction was mainly concentrated in the range of 200–400 g/mol; its heavy average molecular weight increased first and then decreased after incubation. With the increase of incubation time, the substances with molecular weight less than 200 g/mol and more than 400 g/mol gradually decreased, while the components with molecular weight of 200–300 g/mol gradually increased and its molecular weight distribution was more concentrated. This indicated that the component may have undergone polycondensation or cleavage reactions during the incubation period [19].

### 2.3. HI-TS Fraction Variation

Fourier transform infrared spectroscopy (FT-IR) was widely used to identify the structural composition of pitch or to determine its chemical groups. Figure 3 shows the FT-IR spectra of the HI-TS fractions after incubation. The peak appearing near 3420 cm^−1^ was -OH stretching vibrations [21]. Peaks at 1700 cm^−1^ and 1650 cm^−1^ were carbonyl functional group stretching vibrations [22], and the peak at 1030 cm^−1^ was attributed to sulfoxide stretching vibrations [19], while carbonyl and sulfoxide stretching vibrations intensities were enhanced with the increase incubation time. Aliphatic C-H stretching vibrations appeared at 2850 cm^−1^ and 2920 cm^−1^ and aromatic C-H stretching vibration located at 3050 cm^−1^ [23]. The aromatic C-H intensity was higher than that of aliphatic C-H, and the proportion of aliphatic hydrogen for HI-TS-60 increased and then gradually decreased. The aromatic ring C=C stretching vibration was near 1600 cm^−1^ [22], which shifted to higher frequencies and the peak intensity increased in HI-TS-60 and HI-TS-120. This suggested that the intermolecular chemical environment of pitch molecules was changed due to the increase of oxygen-containing functional groups such as hydroxyl, carbonyl, and sulfoxide groups; thus, the polarity of spinnable pitch was promoted. Moreover, the absorption peak of anhydride at 1789 cm^−1^ was also increased. This could be attributed to the intermolecular or intramolecular interaction of hydroxyl and carbonyl groups [24,25].

The ring number distribution of polycyclic aromatic hydrocarbons (PAHs) could be studied using simultaneous fluorescence spectra analysis. Figure 4 shows the fluorescence spectra of HI-TS fraction, where bicyclic aromatic hydrocarbons appeared at 300–340 nm, tricyclic aromatic hydrocarbons were located at 340–400 nm, tetracyclic aromatic hydrocarbons were located at 400–425 nm, and the peak beyond 425 nm was attributed to pentacyclic aromatic hydrocarbons and above [26,27,28]. This suggested that the aromatics in HI-TS fraction were mainly five-ring and more components. It can be seen that the fluorescence spectra intensity of HI-TS-90 between 413 nm and 456 nm was strongest, indicating that the component had the greatest electron conjugation. Table 4 lists the changes in ring numbers at different incubation times, where the dicyclic and tricyclic aromatic hydrocarbons in HI-TS fraction gradually decreased and then increased, while the pentacyclic and more rings aromatic hydrocarbons increased first and then decreased. This indicated that bicyclic and tricyclic aromatic hydrocarbon molecules might be polymerized to form larger aromatic rings in the early incubation period.

The ^13^C-nuclear magnetic resonance (^13^C-NMR) spectra in HI-TS fraction are shown in Figure 5. The chemical shift of 0–90 ppm was attributed to aliphatic carbon and the shift at 100–150 ppm was assigned to aromatic carbon [26]. The chemical shifts of 193 ppm and 60 ppm were a pair of spinning sidebands of the sample during the test. The fitting peaks of the aromatic carbon region are shown in Figure 6. The peaks around 120 ppm and 126 ppm were protonated aromatic carbon, while the peak at 137 ppm was substituted carbon and bridgehead carbon [29,30]. The aromatic carbon percentage of HI-TS-60 was increased to 78.68% (75.11% of HI-TS-0). Moreover, the content of protonated aromatic carbon decreased to 93.67% (96.97% of HI-TS-0). This indicated that the molecules were condensed. As for HI-TS-90, the content of aliphatic carbon (32 ppm) increased, the content of bridgehead carbon and substituted carbon (137 ppm) decreased, and the content of five rings and more decreased. Thus, it was possible that the large aromatic hydrocarbons undergone cleavage and the content decreased, and some of the alkyl side chains were removed during the incubation process. As for HI-TS-150, the content of aromatic carbon increased to 76.56% (71.09% of HI-TS-90), and the content of protonated aromatic carbon decreased to 92.31% (94.61% of HI-TS-90). This indicated that the molecules were condensed after a long incubation period.

The molecular structure evolution of spinnable pitch was consistent with a free radical reaction mechanism [29]. Figure 7 shows the electron paramagnetic resonance (EPR) spectra of HI-TS fraction. The g-value, free radical concentration, and line width of the HI-TS fraction were calculated and are listed in Table 5. It was reported that the g-values of heteroatoms containing unpaired electrons such as N, O, and S were higher than those of aromatic hydrocarbon radicals [31,32]. While most aromatic hydrocarbon radicals had a strong exchange of mutual off-domain π electrons, these radicals were more stable and easily detectable [8,31,33]. Therefore, for HI-TS-60, the unpaired electrons in the heteroatoms may be transferred to the aromatic hydrocarbon [29], and the stable aromatic hydrocarbon radical may result in a large increase in the radical concentration to 7.85 × 10^17^ spins/g (3.81 × 10^17^ spins/g of HI-TS-0). The interaction of these free radicals made molecules condense into macromolecules, and the content of aromatic carbon increased, leading to the increase of the content of five ring and above macromolecules in the early incubation period.

For HI-TS-90, the increase of the g-factor indicated the presence of unpaired electrons on the oxygen atom [34]. The generation of free radicals led to enhance electron-electron spin coupling and widen in linewidth [33,35]. The oxygen-containing radical attacked pentacyclic and more components, and rings were easily opened [36]. Thus, the content of pentacyclic and more components decreased, the aliphatic group detached, and the content increased.

Due to the increase of the incubation time, free radical concentration was increased. The molecules gradually changed from branched laminar to cross-linked and from linear to bulk [25]. It may cause changes in the small molecules of aliphatic structure, with the partial conversion to aromatic carbon. Thus, aromatic carbon content and the di- and tricyclic aromatic hydrocarbons increased. The occurrence of intermolecular cross-linking and the formation of oxygen-containing functional groups restricted the free movement of unpaired electrons, resulting in a decrease in g-value. Moreover, the substantial narrow linewidth of HI-TS-150 may be the exchange effect between aromatic hydrocarbon radicals [31].

In summary, the HI-TS fraction generated various oxygen-containing functional groups, such as hydroxyl, carbonyl, and sulfoxide groups during the incubation process. The content of aromatic carbon in HI-TS fraction was about 70%. In the early stage, molecular condensation occurred, the content of polycyclic aromatic hydrocarbons with five rings and more increased, and the content of two or three-ring substances decreased. Thereafter, pentacyclic and more components may undergo cleavage and their content decreased, while the di- and tricyclic substances subsequently increased. After a longer incubation time, the content of aromatic carbon increased, and the molecules gradually became cross-linked from branched chain laminae.

### 2.4. TI-THFS Fraction Variation

Figure 8 displays the FI-IR spectra of the TI-THFS fraction. The intensity absorption vibration peaks of anhydride, sulfoxide, carbonyl group, and hydroxyl were increased, respectively [19,20,21]. This indicated that this fraction also formed various oxygen-containing functional groups during the incubation period. For TI-THFS-120, the proportion of aliphatic hydrogen increased and the intensity of the absorption peak was higher than that of aromatic hydrogen. The hydroxyl absorption intensity significantly increased in TI-THFS-60, then decreased gradually with the increase of incubation time. The peak of the aromatic ring C=C skeleton stretching vibration shifted to higher frequencies in TI-THFS-60 and TI-THFS-90, while it shifted to lower frequencies in TI-THFS-120 and TI-THFS-150. With the generation of oxygen-containing functional groups, the intermolecular electronic effects also changed, affecting the chemical environment of the C=C backbone of the aromatic ring and causing the absorption peak to shift.

Figure 9 shows the ^13^C-NMR spectra of the TI-THFS fraction of spinnable pitch after different incubation times, where the chemical shift near 125 ppm was aromatic carbon, the shift at 32 ppm was aliphatic carbon [19,29,30], and the sharp peaks in 25 and 67 ppm were the solvent peaks of THF. The fitting peak of aromatic carbon is shown in Figure 10. As for TI-THFS-30 and TI-THFS-60, the aromatic carbon rate increased and the protonated aromatic carbon (near 120 ppm and 126 ppm) content decreased, indicating that the TI-THFS fraction underwent condensation. As for TI-THFS-90, the aromatic carbon content decreased, and the protonated aromatic carbon content in the aromatic carbon decreased by 1% compared with that TI-THFS-60, while the intensity of aliphatic carbon increased. This indicated that condensation of the aromatic fraction continued to generate, and the formation of aliphatic groups resulted in a decrease of aromatic carbon content in the TI-THFS fraction. The aromatic carbon content of TI-THFS-120 increased to 69.2% (65.38% of TI-THFS-90), but the protonated aromatic carbon content in the aromatic carbon increased to 94.16% (92.82% of TI-THFS-90). This indicated that the aromatic carbon content increased and the aliphatic content decreased, but the degree of condensation did not increase. The aromatic carbon rate of TI-THFS-150 increased. This change in the aromatic carbon rate was consistent with that of HI-TS fraction.

Figure 11 shows the fluorescence spectra of the TI-THFS fraction. This suggested that the TI-THFS fraction was mainly composed of tetra- and pentacyclic and more aromatic hydrocarbons. The peaks near 425 nm showed the blue shift and red shift after incubation, while the peak position near 410 nm did not change significantly. This indicated that macromolecular aromatic substances were changed during the incubation process. The peak intensity of TI-THFS-60 was the strongest, while the electron conjugation was the largest [37].

Figure 12 shows the EPR spectra of the TI-THFS fraction. Table 6 lists the g-value, free radical concentration, and linewidth values for the TI-THFS fraction, which has the same g-value at different incubation times, indicating that the type of radicals in the TI-THFS fraction did not change significantly. The relatively low g-value may be due to the induction of the reaction by the more stable aromatic hydrocarbon radicals in the fractions [8,33]. Meanwhile, the peak linewidth was narrowed and the radical concentration was decreased. As for TI-THFS-30 and TI-THFS-60, the content of aromatic carbon increased and the content of protonated aromatic carbon decreased; this indicated that the intermolecular condensation occurred. Thus, the effect of electron-proton coupling could lead to linewidth narrow [31]. During the condensation of the molecule, the alkyl branched chains could be removed, resulting in an increase in the amount of sub-methyl (32 ppm) in TI-THFS-90. As for TI-THFS-120 and TI-THFS-150, the conversion of aliphatic to aromatic hydrocarbons was occurred; thus, the content of aromatic carbon increased. The generation of aromatics leaded to concentrations of free radical and unpaired electrons increased, while the linewidth of EPR peak was widened by the electron-electron coupling effect [31].

The hydroxyl, carbonyl, sulfoxide, and other oxygen-containing functional groups were generated in TI-THFS fraction during the incubation process. Moreover, the stretching vibration of the hydroxyl group was maximum at 60 days of incubation, and then gradually decreased. Most of the PAHs in TI-THFS fraction were tetracyclic and pentacyclic and more components. The components were continuously condensed during the incubation process and some of the aliphatic groups were delocalized. Additionally, with the increase of incubation time, the aliphatic groups were gradually transformed into aromatic hydrocarbons.

### 2.5. THFI Fraction Variation

The FT-IR spectra of the THFI fraction are shown in Figure 13. The peaks at 1600 cm^−1^ and 1450 cm^−1^ were aromatic ring skeleton vibrations [19]. The peaks at 847 cm^−1^, 821 cm^−1^, 748 cm^−1^ were aromatic ring unsaturated C-H out-of-plane deformation vibrations, while the peaks 3046 cm^−1^ were aromatic ring C-H stretching vibrations [20]. This showed that the fraction had a high aromatic component. The stretching vibration of –OH at 3447 cm^−1^ [21] was enhanced in THFI-60 and it was gradually decreased with incubation time. The stretching vibration of the sulfoxide group appeared at 1030 cm^−1^, which gradually increased with the increase of incubation time. Moreover, carbonyl stretching vibration at 1740 cm^−1^ also appeared after a long period of incubation [23]. In summary, the THFI fraction also produced hydroxyl, sulfoxide, carbonyl, and other oxygen-containing functional groups during the incubation process.

Figure 14 shows the ^13^C-NMR spectra of the THFI fraction. The aromatic carbon content of the THFI fraction increased to 76.06% (74.26% of THFI-0). There was a slight increase in the content of aromatic hydrocarbons. The molecule continued to undergo condensation during incubation without any obvious conversion of aliphatic and aromatic carbon.

### 2.6. Melting and Rheological Properties

The melting behavior of the spinnable pitch at different incubation times was observed by the hot stage microscope. Compared with the original spinnable pitch, the initial melting temperature of spinnable pitch incubated for 150 days increased from 185 °C to 215 °C, the complete melting temperature increased from 230 °C to 240 °C, and the softening point increased from 210 °C to 220 °C. The molten morphologies of the spinnable pitch at 250 °C are shown in Figure 15. The melt fluidity of spinnable pitch deteriorated with the increase of incubation time, and some particles appeared on the surface of spinnable pitch after incubation for 150 days. This intuitively expressed that molecular polymerization was formed by intermolecular oxidation, cross-linking, and condensation according to the analysis of above different fraction composition. After 150 days of incubation at 50 °C, the spinnable pitch could be completely melted at 240 °C and its spinnability was still maintained. However, the generation of particles tended to block the spinneret during spinning, greatly reducing the continuity of spinning and making the spinning performance poor.

Figure 16a displays the viscosity-temperature curves of the spinnable pitch after different incubation times. It shows that when the viscosity was above 20,000 cP, higher temperature was required for spinnable pitch to reach the same viscosity with increase incubation time. Figure 16b also shows that the viscosity increased with the increase of incubation time of spinnable pitch at the same shear rate. The variation of polymer viscosity with temperature in a specific range follows the Arrhenius equation [38,39]:(1)$$\mu \left(T\right){=\mathrm{ke}}^{\frac{E}{R}}$$
where k is a constant and E is the displacement factor activation energy, which is proportional to the stress generated during extrusion. The energy aging index (EAI = activation energy change value/original activation energy) was used to assess the degree of asphalt aging [40], where it could be used to evaluate the change in flow activation energy of spinnable pitch before and after incubation. Figure 16c provides double logarithmic plots of viscosity versus temperature at a shear rate of 20 s^−1^. Table 7 lists the flow activation energy and EAI value of the pitch before and after incubation; the flow activation energy of spinnable pitch increased after incubation. The EAI value of R-150 increased dramatically, indicating that large changes occurred within the molecule during incubation. The generated oxygen-containing functional groups such as carbonyl, sulfoxide, and hydroxyl groups could promote the polarity of spinnable pitch, enhance the intermolecular interactions, and increase the shear deformation resistance. Thus, the activation energy of spinnable pitch was increased.

The larger Ea value of the spinnable pitch had lower temperature dependence and thermal sensitivity [40,41], which was ideal for pitch spinning. However, the higher flow activation energy caused an increase in the shear stress of molecule and deteriorated liquidity during melt spinning [38]. Therefore, after a long period of incubation, the increase in viscosity of the pitch did not necessarily have a positive effect on its spinning performance.

## 3. Materials and Methods

### 3.1. Materials

The spinnable pitch was provided by Jiexiu Changlong New Material Technology Co., LTD., China. This spinnable pitch that resulted in air-blown thermal polycondensation was an isotropic pitch. The analytical reagents of n-hexane, toluene, and tetrahydrofuran were purchased from Sinopharm Group (China).

### 3.2. Methods

The spinnable pitch was cracked and placed in 15 grinded flasks with a mass of 20 g; then, these samples were placed in an incubator at 50 °C to accelerate its evolution. These spinnable pitch samples were taken out from the incubator every 30 days, after which it was step-by-step separated into n-hexane, toluene, and tetrahydrofuran using soxhlet extraction; the separation procedure is shown in Figure 1. The corresponding spinnable pitch, n-hexane, toluene, and tetrahydrofuran soluble and insoluble substances are named R-x, HS-x, HI-TS-x, TI-THFS-x and THFI-x, respectively, where x refers to the incubation days. The n-hexane, toluene, and tetrahydrofuran were recovered by rotary evaporation. The n-hexane insoluble, toluene insoluble, and tetrahydrofuran insoluble fractions were dried under vacuum at 70 °C, 110 °C, and 70 °C for 3 h, respectively.

### 3.3. Characterization

An elemental analyzer (Vario EL Cube) was used to determine the contents of carbon, hydrogen, nitrogen, and sulfur in samples. FT-IR spectra were obtained with a Nicolet FT-IR 380 spectrometer (Thermo Electron, USA); the scanning wavelength range is 4000–400 cm^−1^ and the resolution is 4 cm^−1^. NMR analysis was carried out with a Bruker Avance III 600 ^13^C Nuclear Magnetic Resonance (Bruker, Germany). The aromatic carbon ratio (*f_a_*) of the sample was calculated by integrating different chemical shifts according to the integral area ratio of aromatic carbon to total carbon [22]. The concentration and type of stable free radicals in the samples were derived from EPR spectra (EMXPLUS 10/12, Bruker, Germany). The free radical concentration (R_c_) and g-value were calculated by $R_{c}{=N}_{R}/m$ [34] and ${g=h\upsilon /\beta B}_{r}$ (N_R_ and m are the numbers of free radicals (spins) and mass (g) of the sample, h is the Planck’s constant, υ is the microwave frequency, β is the Bohr magnetron, and B_r_ is the resonant magnetic induction) [35], respectively. The polycyclic aromatic hydrocarbons ring number of spinnable pitch was evaluated using a HITACHI F7000 UV-fluorescence spectrometer, where the scanning rate of the spectrometer is 240 nm/min, the scanning range is 240–600 nm, and the gap width is 5 nm. Matrix-assisted laser desorption ionization-time of flight mass spectrometry is suitable for the determination of molecular weight and the distribution of samples was determined by an ultra flex MALDI-TOF (Bruker, Germany). The percentage and distribution of molecular weight in the fractions was determined by GPC using an agilent 1260 (Germany). The chromatographic columns were Water Syragel HR 0.5 and HR 1 THF columns, where THF was the mobile phase, the flow rate was 0.8 mL/min, the injection concentration was 0.1 mg/mL, and the injection volume was 20 μL. The softening point and melting process of the spinnable pitch were tested and observed using an RT600 precision hot stage equipped with an MM25 microscope. The process was carried out at a temperature rise rate of 10 °C/min with an argon flow of 20 mL/min. The rheological properties of the spinnable pitch were measured by a Brinell viscometer SNB-AI under nitrogen atmosphere.

## 4. Conclusions

In summary, the HS, HI-TS, TI-THFS, and THFI fractions evolution of spinnable pitch at 50 °C for 150 days was investigated and analyzed. The HS fraction may undergo condensation or cleavage during incubation, and the molecules tended to develop in the range of 200–300 g/mol. HI-TS, TI-THFS, and THFI fractions all generated oxygen-containing functional groups, such as hydroxyl, carbonyl, sulfoxide, and anhydride, during incubation. The HI-TS fraction occurred molecular condensation and cross-linking in the presence of oxygen-containing radicals and aromatic hydrocarbon radicals. Some of the aliphatic groups may be converted to aromatic hydrocarbons during incubation and the content of aromatic carbon increased. TI-THFS fraction tended to change in large molecules of PAHs. The aliphatic group may be detached during the molecular condensation process. Moreover, after a long period of incubation, molecules were condensed, and the rate of aromatic carbon increased. THFI fraction had high aromaticity, molecules were condensed during the incubation process, and the content of aromatic carbon increased. These change of composition and structure of spinnable pitch led to its softening point, increasing viscosity and flow activation energy, and deteriorating rheological property, which further affected its spinning behavior and pitch fiber performance, and finally resulting in pitch carbon fiber quality degradation.

## Data Availability

The data presented in this study are available on request from the corresponding author.

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
