# Peer review of "Evolution of the Composition and Melting Behavior of Spinnable Pitch during Incubation"

_molecules, 2023, doi:10.3390/molecules28031097_

Round 1

Reviewer 1 Report

I believe that the manuscript (moleculars-2150945) has a good topic to understand details about the change of four chemical component in spinnable pitch during storage, Yet, to publish in the journal of Moleculars, the manuscript needs to carefully modify, including format and some important data

This manuscript (moleculars-2150945) reports the composition evolution of spinnable pitch during a long storage time. I have some comments.

1.    Please carefully recheck the format, spelling and other issues, such as: TS-THFS at line 102, the format of Figure11, the spelling of abscissa in Figure 16a, THFIS-x at line 372, ml/min at line 400……

2.    hot table microscope at line 319 should be replaced by hot stage microscope.

3.    the corresponding drying temperature and leaching time should be provided in Methods section.

4.    “The stretching vibration of the sulfoxide group appeared at 1030 cm-1, it gradually increased with the increase of storage time.” It seems that the peak at 1030 cm-1 is not obvious in Figure 13.

5.    As we all known, the physical-chemical properties of pitch varied from the precursor resource and synthesis method, please provide the preparation method of the spinnable pitch and property of raw material.

The work is novel and interesting, can be published after minor revision.

Reviewer 2 Report

The manuscript explained the physical and chemical properties of the pitches obtained by different extraction processes then verified change of the properties with incubations, resulting in the rheological property change and ultimately estimating the spinning conditions. As stated in the manuscript, unlike many previous studies that mainly focused on thermal behavior confirmation, the manuscript performed various analyses to figure out physical and chemical properties of the pitch extractions with incubations; thus this manuscript showed a unique benefit to publish in this journal. I believe this study would be useful to understand spinnable pitch compositions, providing suitable continuous melt-spinning conditions. The manuscript offered sufficient results to support the authors` claim and the claims seem to be convincing, however, some parts should be modified before publication like below.

1.       I understand that Scheme 1 may be put in 3.2. Methods section. For the readers, however, I prefer to put Scheme 1 prior to the 2. Results and discussions section.

2.       The manuscript didn`t include pitch spinning results, hence the author should change the title; the modified title should include what the authors mainly did.

3.       All the pitches were not “storage” but “incubation” with heating. Thus, the authors may change the “storage” to “incubation”.

4.       The authors suggested some examples as coal tar pitch-based functional carbon materials; however, there is no reference in there (lines 32-34). Some references, including below, should be attached to show the right applications

                                     i.              https://www.nature.com/articles/s41598-017-05192-5

                                   ii.              https://www.sciencedirect.com/science/article/pii/S0008622315300531

                                 iii.              https://journals.sagepub.com/doi/10.1177/1528083718763774

                                 iv.              https://onlinelibrary.wiley.com/doi/full/10.1002/adem.202001523

5.       If possible, the authors provide thermal properties of the pitches with incubations, such as a softening point, to figure out the spinning possibility. In addition, the authors can show that the used spinnable pitch is an isotropic or mesophase-based one.

6.       It is hard to understand Figure 15. Is there any supporting data to strengthen the authors` opinion? ie. Surface profiler to figure out the smoothness. Also, can you provide where the particles were derived from? If the particles are impurities (not pitch derivatives), the particles should be removed by filtering or purification before the spinning process. The authors can determine the particle elements by EDS easily. If the particles were derived from pitch agglomeration having higher melting behavior, please provide reliable results to support the authors` intuitive claim.

7.       Please specify 3.2. Methods. The 20g of the spinnable pitch was detached from a bulk pitch every 30 days? or separated 20g pitches kept in the incubator? The detaching process may cause impurity attachment on the bulk pitch surface, leading to inaccurate results.

8.       Many abbreviations are missing in the manuscript.

9.       Line 16: “was” should be “were”.

10.    Line 96: “0 to150” should be “0 to 150”.

11. Modify Figure 13 by baseline correction
